# Random Time Division Multiplexing Based MIMO Radar Processing with Tensor Completion Approach

**DOI:** 10.3390/s23104756

**Published:** 2023-05-15

**Authors:** Yuan Zhang, Yixue Qiao, Gang Li, Wei Li, Qing Tian

**Affiliations:** 1School of Information, North China University of Technology, Beijing 100144, China; qiaoyix@126.com (Y.Q.); lwsar@ncut.edu.cn (W.L.); tianqing@ncut.edu.cn (Q.T.); 2School of Information Science and Technology, Tsinghua University, Beijing 100084, China; gangli@tsinghua.edu.cn

**Keywords:** automotive radar, multiple-input–multiple-output (MIMO), random time division multiplexing (Random TDM), tensor completion

## Abstract

Automotive radar pursues low cost and high performance, and especially hopes to improve the angular resolution under the condition of a limited number of multiple-input–multiple-output (MIMO) radar channels. Conventional time division multiplexing (TDM) MIMO technology has a limited ability to improve the angular resolution without increasing the number of channels. In this paper, a random time division multiplexing MIMO radar is proposed. First, the non-uniform linear array (NULA) and random time division transmission mechanism are combined in the MIMO system, and then a three-order sparse receiving tensor of a range-virtual aperture-pulse sequence is obtained during echo receiving. Next, this sparse three-order receiving tensor is recovered by using tensor completion technology. Finally, the range, velocity and angle measurements are completed for the recovered three-order receiving tensor signals. The effectiveness of this method is verified via simulations.

## 1. Introduction

In recent years, millimeter-wave (MMW) radar has been widely used in automatic driving systems to obtain various information, such as the range, angle, speed and target scattering coefficient or target point cloud contour of the road environment, which gives the automatic driving system greater perception capabilities [1]. Among them, the acquisition of the range depends on transmitting a high-bandwidth chirp signal. After dechirp processing has been performed on the echo, the frequency domain response of the target in the slant-range direction can be obtained via Fourier transform. The acquisition of velocity depends on transmitting multiple chirp signals continuously from the transmitting antenna, which determines the target Doppler domain response via Fourier transform. The acquisition of the angle depends on the phase difference in the target wavefront between multiple physical apertures, and the spatial directional response of the target aperture can be obtained via the spectral analysis method [2].

In order to obtain the above parameters, the millimeter-wave radar system is usually implemented according to a certain data processing process. First, the two-dimensional Fourier transform is performed on the echo data of each receiving channel in the range and Doppler directions in order to obtain the target response in the range-Doppler domain. The involved studies here are Doppler estimation [3], Doppler ambiguity resolution [4,5] and the problem of range migration correction, caused by target velocity [6]. Then, the millimeter-wave radar system uses the target detection method, such as the commonly used constant false alarm detection algorithm, to obtain the position of the target in the range-Doppler domain. Finally, the detected complex scattering signals of the target in the range-Doppler domain of each channel form the receiving vector, from which the target angle information can be obtained by using the spectral analysis method [7]. Although the above processing flow is relatively complete and fixed, it still faces many challenges [8], including how to improve the orthogonality between transmit channels via waveform coding [9,10,11,12,13], which can greatly reduce the mutual interference [14]. In addition, there are other problems worth investigating, e.g., target sidelobe suppression [15] or multi-channel amplitude and phase error calibration [16].

In the actual millimeter-wave radar system, the issue of angular resolution improvement has received a lot of attention. The angular resolution is directly related to the target recognition and point cloud density. Automotive radar has been pursued in order to obtain a high angular resolution as much as possible under the condition of limited hardware resources. The angular resolution is theoretically determined by the length of the physical aperture of radar. Therefore, improving the angular resolution means that the number of radar apertures needs to be increased. However, it is unfeasible to add more real physical apertures to improve the angular resolution for the profit-seeking trend of automotive radar. The current mainstream approach is to use MIMO technology that utilizes virtual apertures to improve the angular resolution. For example, the commonly used three-transmitting and four-receiving (3T4R) millimeter-wave chip can provide 12 virtual apertures, which is equivalent to an angular resolution of about 9.5 degrees under the condition of a uniform linear antenna array (ULA) spaced at half wavelengths. To further improve the angular resolution, a non-uniform linear antenna array (NULA) can be used, that is, the aperture spacing is no longer a uniform half-wavelength, but a random non-uniform distribution [17]. In [18], the researchers use the compressed sensing theory to improve the minimum number of transmit and receive antennas for the NULA, which are proportional to the number of targets and the logarithmic dependence on the array aperture. Reference [19] further combines the matrix completion technology to suppress the high sidelobe induced by the sub-Nyquist sampling of the NULA on the basis of a non-uniform array.

Currently, the development of automotive radar is constrained by both cost and volume, so blindly increasing the number of MIMOs is not feasible in reality. Under the premise of not increasing the number of MIMOs, in order to improve the angular resolution, the MIMO must be sparsely distributed, thereby increasing the aperture. The problem brought about by this is determining how the data at the missing aperture can be filled in, so when the aperture is missing, aperture completion becomes a key issue. If the conventional time division multiplexing signal transmission and reception mechanism is still used, the aperture data obtained by MIMO has a single distribution over time. Therefore, we imagine that if the distribution mode is randomized, it will help the recovery of missing aperture data.

In this paper, a NULA is adopted, and a random time division multiplexing (TDM) transmission mechanism is introduced into each transmitting antenna to form a sparse three-order receiving tensor of a range-virtual aperture-pulse sequence. Then, the tensor completion technique [20] is used to realize the filling of this three-order receiving tensor for recovery. Finally, the range, velocity and angle measurement are obtained using the recovered tensor. Simulations show that, compared with the ULA, the proposed method can improve the angular resolution under the condition that there are the same transmission and reception channels. The difference between the working mode of this paper and the conventional TDM MIMO is shown in Figure 1. Taking 3T4R MIMO as an example, in the conventional TDM mode, the system transmits signals in sequence according to the order of transmit antennas T1, T2, and T3, and periodically cycles accordingly. Unlike the regular process, the transmit order of the antennas is random in this paper. Moreover, we combine the NULA and random TDM to forma sparse receiving three-order tensor signal as shown in Figure 1b. It should be noted that the non-uniform distribution and transmission timing of the virtual receiving channels in Figure 1b are completely random, and the figure here only illustrates one specific case.

## 2. Signal Model

In this section, the signal model for the random time division multiplexing is given. The geometric diagram of the target detection using the NULA MIMO is shown in Figure 2. This assumes that the target echo is in the form of a plane wave, and that the target scattering coefficient is a unit value. The distance between the target and the *n*-th virtual channel is as follows [19]:(1)$$r_{p}=r_{0}+\frac{1}{2}d_{n}\cdot \sin\theta +\frac{1}{2}m\cdot v\cdot T_{p}$$
where *r*_0_ is the distance between the target and the 0-th virtual receiving channel, *d_n_* is the position of the *n*-th virtual receiving channel on the coordinate axis OX, *θ* is the incident angle of the target echo, *m* represents the number of signal sequences transmitted by the current channel, *v* is the relative velocity in the line-of-sight direction between the target and the radar, and *T_p_* is the period of the transmitted signal.

The transmitted signal is as follows [21]:(2)$$s(\tau )=W_{r}(\tau )\cdot \exp\left\{j\cdot \left(2\pi f_{c}\tau +\pi k_{r}\tau^{2}\right)\right\}$$
where *j* is the imaginary unit, *τ* is the fast time, which means the time history of a pulse from transmission to reception [19], *W_r_*(·) is window function of the chirp signal, *f_c_* is the carrier frequency, and *k_r_* is the chirp rate of the transmitted signal. The slant-range of the point target *P* expressed by (1) induces the time delay of the transmitted signal, which is shown as follows:(3)$$\tau_{p}=\frac{r_{p}}{c}$$
where *c* is the velocity of light.

Therefore, the echo signal is expressed as follows [21]:(4)$$s\left(n,m,\tau \right)=W_{a}(n)\cdot W_{v}(m)\cdot W_{r}(\tau -\tau_{p})\cdot \exp\left\{j\cdot \left[2\pi f_{c}\left(\tau -\tau_{p}\right)+\pi k_{r}{\left(\tau -\tau_{p}\right)}^{2}\right]\right\}$$
where *W_a_*(·) and *W_v_*(·) are the window functions in the direction of the virtual receiving channel and signal sequence per channel, respectively. The received echo data have three dimensions, that is, the range direction, the virtual receiving channel direction and the signal sequence direction. It should be noted that the channel spacing is non-uniformly distributed here, and the signal sequence is also random.

Millimeter-wave radars usually use the dechirp technique to reduce the sampling requirements by mixing the received and reference signals. The reference signal is given by the following:(5)$$s_{c}=\exp\left\{j\left[2\pi f_{c}\left(\tau -\tau_{c}\right)+\pi k_{r}{\left(\tau -\tau_{c}\right)}^{2}\right]\right\}$$
where *τ_c_* = 2*r_c_*/*c* is the time delay of the reference signal, and *r_c_* is the reference slant-range. The dechirped signal can be expressed as follows [22]:(6)$$s_{IF}\left(n,m,\tau \right)=s\left(n,m,\tau \right)\cdot s_{c}^{*}=W_{a}(n)\cdot W_{v}(m)\cdot W_{r}(\tau_{d}-\tau_{\Delta })\;\cdot \exp\left\{j\cdot \left[-2\pi f_{c}\tau_{\Delta }-2\pi k_{r}\tau_{d}\tau_{\Delta }\right]\right\}\cdot \exp\left\{-j\cdot \left[\pi k_{r}\tau_{\Delta }^{2}\right]\right\}$$
where $\tau_{d}=\tau -\tau_{c}$ and $\tau_{\Delta }=\tau_{p}-\tau_{c}$. * represents the complex conjugate.

The last exponential term in (6) is well known as the residual video phase (RVP). It can be eliminated by using the ‘Deskew’ method [23]. After the RVP removal, we obtain
(7)$$s_{IF}\left(n,m,\tau_{d}\right)=W_{a}(n)\cdot W_{v}(m)\cdot W_{r}(\tau_{d}-\tau_{\Delta })\cdot \exp\left\{j\cdot \left[-2\pi f_{c}\tau_{\Delta }-2\pi k_{r}\tau_{d}\tau_{\Delta }\right]\right\}$$

Performing the time–frequency substitution of $f_{r}=k_{r}\tau_{d}$ yields
(8)$$S\left(n,m,f_{r}\right)=W_{a}(n)\cdot W_{v}(m)\cdot W_{r}(\frac{f_{r}}{B_{r}})\cdot \exp\left\{j\cdot \left[-2\pi \left(f_{c}+f_{r}\right)\tau_{\Delta }\right]\right\}$$
where *B_r_* is the bandwidth of the transmitted signal.

Next, we substitute (3) into (8) to obtain the following:(9)$$S\left(n,m,f_{r}\right)=W_{a}(n)\cdot W_{v}(m)\cdot W_{r}(\frac{f_{r}}{B_{r}})\cdot \exp\left\{j\cdot \left[-\frac{4\pi }{c}\left(f_{c}+f_{r}\right)\cdot \left(r_{p}-r_{c}\right)\right]\right\}$$

We define
(10)$$K=\frac{f_{c}+f_{r}}{c/2}$$
where *K* is the wavenumber that represents the phase cycles per unit distance in the range direction. Then, (9) can be rewritten as follows:(11)$$S\left(n,m,K\right)=W_{a}(n)\cdot W_{v}(m)\cdot W_{r}(K)\cdot \exp\left\{j\left[-2\pi K\cdot \left(r_{p}-r_{c}\right)\right]\right\}$$
where *W_r_*(*K*) is the envelope function in the range wavenumber direction.

Normally, the system will set *r_c_* to 0, even if it is not 0, and can be compensated by this known quantity, as follows:(12)$$S\left(n,m,K\right)=S\left(n,m,K\right)\cdot \exp\left\{j\left[-2\pi K\cdot r_{c}\right]\right\}=W_{a}(n)\cdot W_{v}(m)\cdot W_{r}(K)\cdot \exp\left\{j\left[-2\pi K\cdot r_{p}\right]\right\}$$

Next, we substitute (1) into (12) to obtain the following:(13)$$S\left(n,m,K\right)=W_{a}(n)\cdot W_{v}(m)\cdot W_{r}(K)\cdot \exp\left\{j\left[-2\pi K\cdot \left(r_{0}+\frac{1}{2}d_{n}\cdot \sin\theta +\frac{1}{2}m\cdot v\cdot T_{p}\right)\right]\right\}$$

From (13), we can see that the received signal phase ultimately consists of three terms. The first term is the linear phase determined by the target distance *r*_0_, which decides the range measuring. The second term is the linear phase determined by the multiple virtual receiving channels, which affects the direction of arrival (DOA). The third term is determined by the target motion, which serves as the velocity estimation. This received signal is a three-order tensor with three dimensions (3-D) that represent range, spatial (virtual aperture) and Doppler (pulse sequence), respectively. In a conventional system, the Fourier transform is performed on the received echo along the range direction in order to realize the range. Meanwhile, it performs the Fourier transform along the Doppler direction to complete the speed measurement. After that, the spectral analysis is made in the spatial direction to complete the angle measurement. However, due to the use of random emission and the NULA, the three-order tensor signals in this paper are randomly and non-uniformly distributed in the Doppler and the spatial direction, resulting in the characteristic of sparsity that cannot be directly processed using conventional methods. However, the advantages brought by this sparse three-order tensor are also obvious. Compared with conventional MIMO, on one hand, the length of the aperture formed in the spatial direction is larger, which means that it can generate a higher angular resolution. On the other hand, combined with random emission in the Doppler direction, tensor completion is used to make up for the sparsity problem as the MIMO channel spacing becomes larger.

## 3. Random TDM MIMO Signal Processing with Tensor Completion

A tensor is the generalization of a matrix to high dimensions. For a tensor *S*∈ℂI1×⋯×IN, the number of dimensions *N* is called the order, also known as the way or mode. In the paper, the *i*-th entry of a vector *s* is denoted by *s_i_*, and the (*i*,*ς*)-element of a matrix **S** is denoted by *s_i,ς_*, in which *i* and *ς* are the row and column indices of the matrix, respectively. In addition, the (*i*_1_,*i*_2_,…,*i_N_*)-element of an order-*N* tensor *S* is denoted by si1,i2,⋯,iN. For more details about the tensor, please refer to [24]. From (13), the echo data in this paper are a three-order tensor *S*(*n*,*m*,*K*), which is represented by *S* in the following, and *N* = 3.

We aim to recover *S* from its small subset of entries $P_{\Omega }\left(S\right)$, as expressed by
(14)$${\left[P_{\Omega }\left(S\right)\right]}_{i_{1},i_{2},i_{3}}=\left\{\begin{array}{c} s_{i_{1},i_{2},i_{3}}\\ 0 \end{array}\right.\begin{array}{c} \mathrm{if}\;\left(i_{1},i_{2},i_{3}\right)\in \Omega \\ \mathrm{otherwise} \end{array}$$

We build the following reconstruction model to obtain S^ as the completion of *S*,
(15)minU(n)∑r=1R∑n=13ℝℚrU(n)∗+η2PΩS−PΩS^
where ℚr extracts the *r*-th column from matrix U(n) for *r* = 1, ⋯, *R*, and *n* = 1, 2, 3. ⋅∗ denotes the matrix nuclear norm and *η* is a regularization parameter that trades off the nuclear norm against the data consistency. ℝ:ℂIn→ℂD1(n)×D2(n) is a linear operator that transforms the vector into a Hankel matrix for integers D1(n) and D2(n), thus satisfying D1(n)+D2(n)=In+1, as follows:(16)ℝar(n)k,l=ar(n)k+l−1∀1≤k≤D1(n),1≤l≤D2(n)

The tensor decomposition is represented as follows:(17)$$X=\sum_{r=1}^{R}u_{r}^{\left(1\right)}\circ u_{r}^{\left(2\right)}\circ u_{r}^{\left(3\right)}=[[U^{\left(1\right)},U^{\left(2\right)},U^{\left(3\right)}]]$$
where *R* is the estimated rank of the reconstructed tensor, the symbol ◦ denotes the vector outer product and the vector u^(*n*)^; for all 1 ≤ *n* ≤ *N*, this is called a *factor*. [[]] is the Tucker operator [25].

The augmented Lagrangian function of (15) is as follows:(18)Lβ(U,Z,D)=∑r=1R∑n=13Dr(n),ℛQrU(n)−Zr(n)+||Zr(n)||∗+β2||ℛQrU(n)−Zr(n)||F2+λ2||PΩ(Y)−PΩ(U(1),U(2),…,U(N))||F2
where Dr(n) is the matrix of Lagrange multipliers. Zr(n)=ℝℚr U(n).

If $\underset{k\to \infty }{\lim}||D_{r;k+1}^{{}^{\left(n\right)}}-D_{r;k}^{{}^{\left(n\right)}}|{|}_{F}=0$, where *k* is the iteration step, then the limit of $\left\{U_{k}\right\}$ satisfies the KKT condition of (15).

The optimization model described in (15) can be solved via the alternate direction method of the multipliers-based algorithm proposed in [20], which exploits the exponential structure of factor vectors. Figure 3 shows a schematic diagram of the completion of the “hole” filling. T1, T2 and T3 represent signals from transmitting channels 1, 2 and 3, respectively. The virtual receiving channels are arranged along the *n* direction. The transmitting cycle time is arranged along the *m* direction. The *K* direction represents pulse sampling points. Data *S* is the data after normal receiving. Since we adopt the NULA layout and random transmission, the signal at the vacancy, i.e., the no signal area, needs to be filled. Using the method in this paper, data *X* are expected to be obtained, that is, the signal of the vacancy is to be restored. After tensor completion, we will continue to use conventional ranging, speed and angle measurement methods for *X* in (17).

First, we perform the inverse Fourier transform on (17) along the range wavenumber *K* direction to obtain the following:(19)$$sss(n,m,\tau )=W_{a}(n)\cdot W_{v}(m)\cdot \delta_{r}(\tau -2r_{p}/c)\cdot \exp\left\{j\cdot \left[4\pi /c\cdot f_{c}\cdot r_{p}\right]\right\}$$
where $\delta_{r}$(·) is the range spatial ambiguity function. Under the assumption that the range migration does not exceed the range resolution unit, (19) is approximately equal to the following:(20)$$sss(n,m,\tau )\approx W_{a}(n)\cdot W_{v}(m)\cdot \delta_{r}(\tau -2r_{0}/c)\cdot \exp\left\{j\cdot \left[4\pi /c\cdot f_{c}\cdot r_{p}\right]\right\}\;=W_{a}(n)\cdot W_{v}(m)\cdot \delta_{r}(\tau -2r_{0}/c)\cdot \exp\left\{j\cdot \left[4\pi /\lambda \cdot \left(r_{0}+\frac{1}{2}d_{n}\cdot \sin\theta +\frac{1}{2}m\cdot v\cdot T_{r}\right)\right]\right\}$$

Then, the Fourier transform is performed along the direction of the pulse sequence *m* to obtain the following:(21)$$sSs(n,f_{m},\tau )=W_{a}(n)\cdot \delta_{v}(f_{m}-\frac{v}{\lambda })\cdot \delta_{r}(\tau -2r_{0}/c)\;\cdot \exp\left\{j\cdot \left[4\pi /\lambda \cdot \left(r_{0}+\frac{1}{2}d_{n}\cdot \sin\theta \right)\right]\right\}$$
where $\delta_{v}$(·) is the Doppler ambiguity function. After the processing of (20) and (21), the range-Doppler image map of each virtual channel can be obtained, and then the target can be extracted from this range-Doppler image map via the detection algorithm [2]. Next, the complex data in the range-Doppler image of each channel where the target is located form the receiving vector in order to further realize the angle measurement, as follows:(22)$$SSs(k_{n},f_{m},\tau )=\delta_{a}(k_{n}+\frac{\sin\theta }{\lambda })\cdot \delta_{v}(f_{m}-\frac{v}{\lambda })\cdot \delta_{r}(\tau -2r_{0}/c)\cdot \exp\left\{j\cdot \left[4\pi /\lambda \cdot r_{0}\right]\right\}$$
where $\delta_{a}$(·) is the angular ambiguity function.

Finally, the target is focused to a point in the range, angle and Doppler direction. Therefore, the range, angle and velocity information of the target can be obtained. The data processing flow of this paper is summarized in Figure 4.

## 4. Performance Simulation and Analysis

In this section, we use the parameters shown in Table 1 for simulation verification. In automotive radar, the mainstream uses 77 GHz as the carrier frequency, and the other parameters we use follow the common configuration used in the industry as much as possible.

### 4.1. Random TDM MIMO Radar Simulation

A simulation scene is built according to Figure 5, where we arrange a 3 × 3 square lattice of point targets that are spaced 15 m in the X direction and 5 m in Y direction, respectively. In addition, the center point target is 85 m away from the MIMO. The used physical array layout of MIMO is shown in Figure 6. We use a six-transmitting and eight-receiving (6T8R) MIMO radar. The transmitting antennas are uniformly distributed with an interval of 5λ, while the receiving antennas are non-uniformly distributed with a minimum interval of 0.5λ. With this antenna layout, the system has a nominal angular resolution of 1.9° and a nominal range resolution of 0.5 m.

After the simulation, the target imaging results are shown in Figure 7a with nine point targets clearly visible. As a comparison, we give the results of the 6T8R in the TDM ULA mode, as shown in Figure 7b, in which the antenna arrays are uniformly distributed with a minimum spacing of 4λ for the transmitting antennas and a minimum spacing of 0.5λ for the receiving antennas; this means that there is a nominal angular resolution of 2.4°. We use the conventional processing method for the ULA echo in Figure 7b, that is, after the echo data is received, the Fourier transforms are directly performed in the range and angle directions.

Next, we analyze the focus quality of the center point in the scene, as shown in Figure 8. The angular response of the −3 dB width using this method is 1.9°, and the angular response of the −3 dB width of the 6T8R MIMO in the TDM ULA mode is 2.4°. It can be seen that the method in this paper definitely improves the angular resolution without increasing the number of antennas. The comparison result of the range response is also given in Figure 9, with a −3 dB width of 0.5 m. Therefore, the simulation results of this method reach the nominal resolution. In reality, the higher sidelobe is indeed disadvantageous and may interfere with weak targets. Because the TDM NULA uses a non-uniform array, the side lobe is higher than that of the TDM ULA, which is one of its limitations.

The purpose of this paper was to improve, compared to the ULA, the angular resolution by using the non-uniform array under the condition of the same transmission and the same reception channels; in addition, it is desirable to reduce the hardware costs as much as possible in practical application. Therefore, the configuration of the NULA transmit channel spacing of 5λ is adopted in this section to form a non-uniform array in the condition of the 6T8R. If the ULA has the same transmission interval, the number of receiving antennas will be increased, that is, the configuration of 6T10R will be reached. As shown in Figure 10, we made such a comparison. It can be seen that even if the ULA has more receiving channels than the NULA, the angular resolution of the two is equivalent in the end, which shows from another perspective that under the same transceiver configuration conditions, the method in this paper has the ability to improve angular resolution. However, as the transmit channel spacing increases, the performance of the NULA will be lower than that of the ULA, as shown in Table 2, mainly because the ULA is a full array.

In real scenarios, the angular resolution of objects (van and motorcycles) with different radar cross-sections (RCSs) is required. The RCS of a van is about 10 dB higher than that of a motorcycle [26]. Therefore, we set two targets at the same distance and angle from the radar, and their RCSs differ by 10 dB. The resolution comparison results are shown in Figure 11. It can be seen that the main lobe responses of the two targets are consistent, but the sidelobes of the larger RCS target are higher.

Finally, for the velocity measurement, the Monte Carlo simulations are performed. In addition, the relationship between the SNR and the root-mean-square error (RMSE) of the velocity measurement is shown in Figure 12. It can be seen that the velocity measurement is more accurate when the SNR is greater than 5 dB.

### 4.2. Performance Analysis of Angular Resolution Improvement

In this subsection, we analyze the ability of the random TDM NULA mode to improve the angular resolution. We still follow the 6T8R MIMO pattern in subsection A, but keep increasing the transmit antenna separation distance from 11 to 16 times at a half wavelength spacing, which corresponds to an change in the angular resolution from 1.8° to 1.2°. The processing results are shown in Figure 13 with different angular resolution processed results from Figure 13a–e. It can be seen that the target response resolution is continuously improved, but the peak-to-sidelobe ratio (PSLR) gets worse and worse. The PSLR also reflects the focusing quality of the target. In automotive radar, a high PSLR is beneficial to the identification of weakly scattered targets such as pedestrians. Combined with the data in the previous subsection, we give the statistical results of the target PSLR when there is a change in the angular resolution in Table 2, and plot the PSLR and angular resolution curves in Figure 14, which can be used as a base reference for balancing the PSLR and angular resolution in practice. As a comparison, we also give the results of the ULA in Table 2. It can be seen that with the increase in the transmission channel spacing, although the NULA is barely comparable to the ULA in terms of angular resolution, the performance of the PSLR will be lower than that of the ULA.

## 5. Conclusions

In this paper, the working mechanism of MIMO radar is improved. We propose a MIMO radar with a non-uniform linear antenna array layout that performs a random time division multiplexing operation in order to form a three-order sparse receiving tensor signal in the range–Doppler–angle direction. Using tensor completion technology, we recover the full signal of the three-order tensor. On this basis, we further obtain range, speed and angle measurements. Compared with conventional MIMO radar processing, the method used in this paper is beneficial to the NULA layout design and significantly improves, compared with the ULA, the angular resolution under the condition of the same transmission and receiving channels. The conventional 6T8R MIMO angular resolution can be improved from 2.4° to about 1°. In addition, when the SNR is greater than 5 dB, the performance of the speed measurement performance is good. This method helps to maximize the system performance potential without increasing the chip channel count.

## Figures and Tables

**Figure 1 sensors-23-04756-f001:**
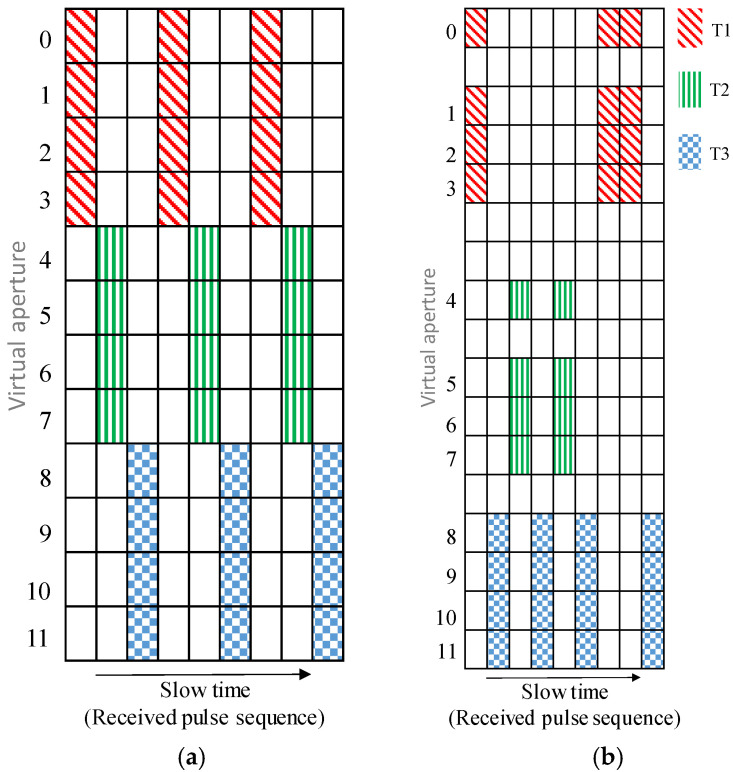
Schematic diagram comparison between conventional and proposed random TDM; (**a**) Conventional TDM, (**b**) Random TDM.

**Figure 2 sensors-23-04756-f002:**
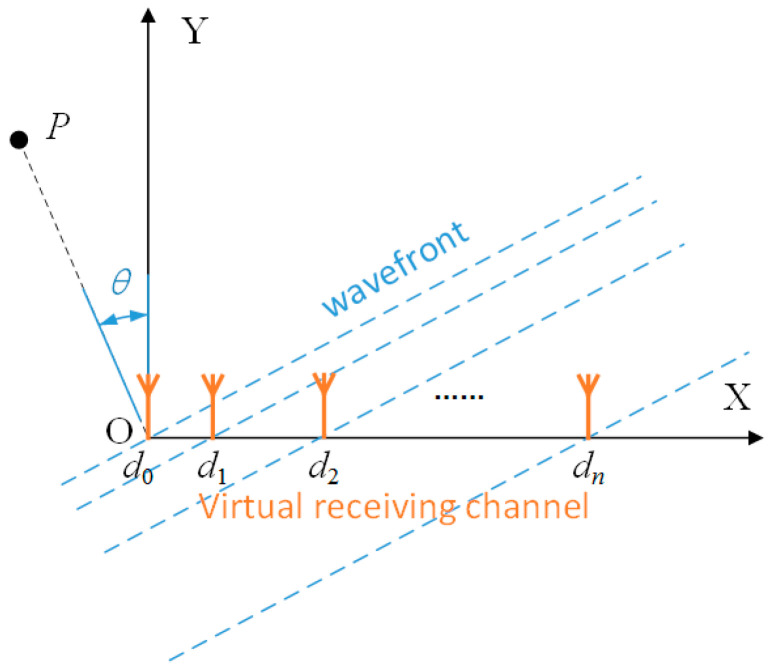
Schematic diagram of target detection geometry using the NULA MIMO.

**Figure 3 sensors-23-04756-f003:**
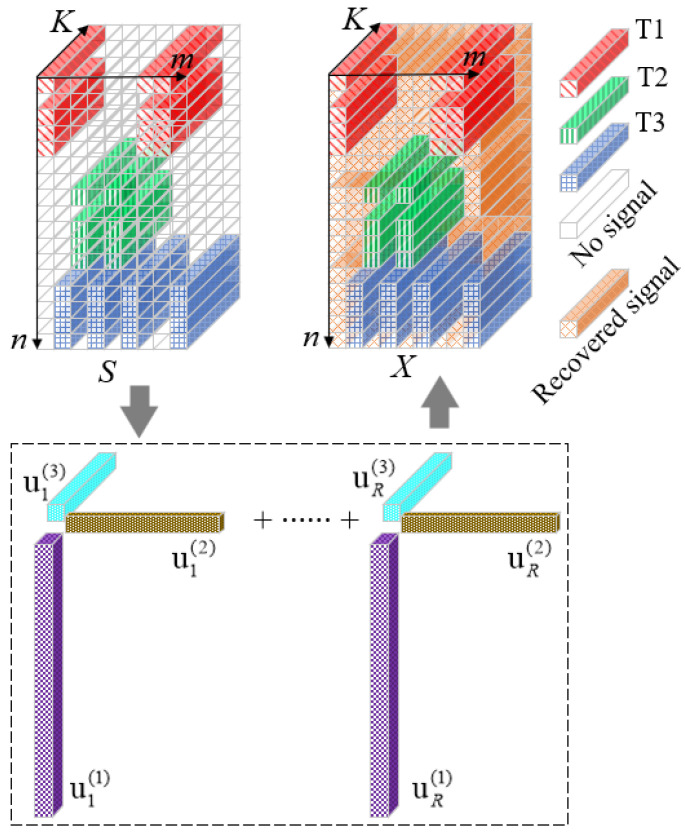
Schematic diagram of receiving echo through tensor completion.

**Figure 4 sensors-23-04756-f004:**
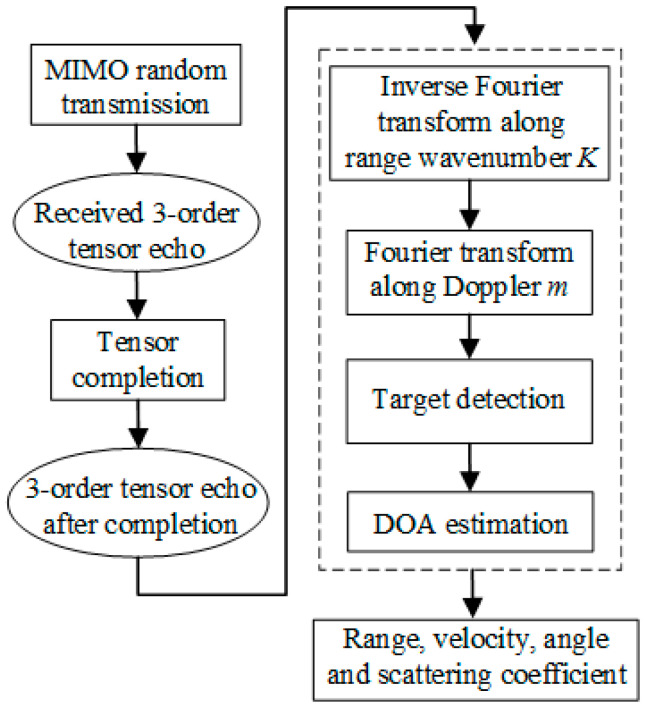
The data processing flow chart of this paper.

**Figure 5 sensors-23-04756-f005:**
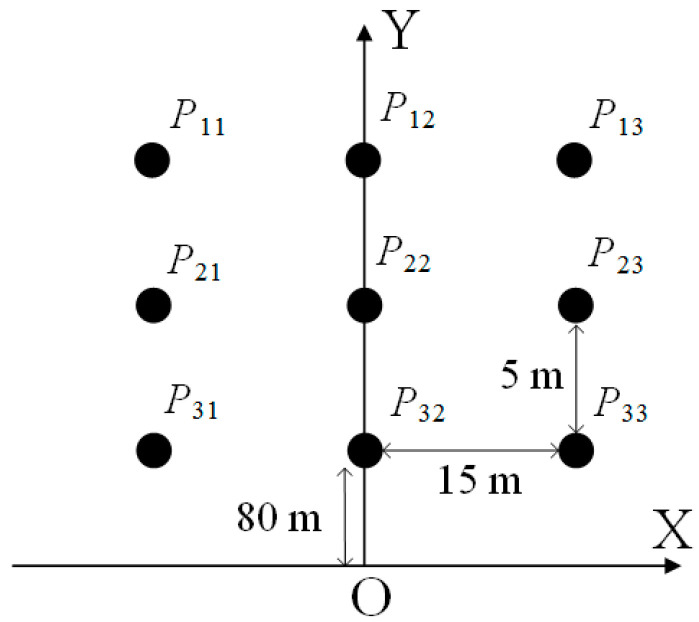
The lattice layout of simulation point targets.

**Figure 6 sensors-23-04756-f006:**
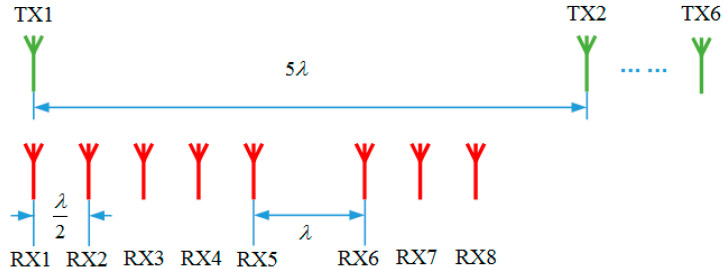
The physical array of the 6T8R MIMO for simulation.

**Figure 7 sensors-23-04756-f007:**
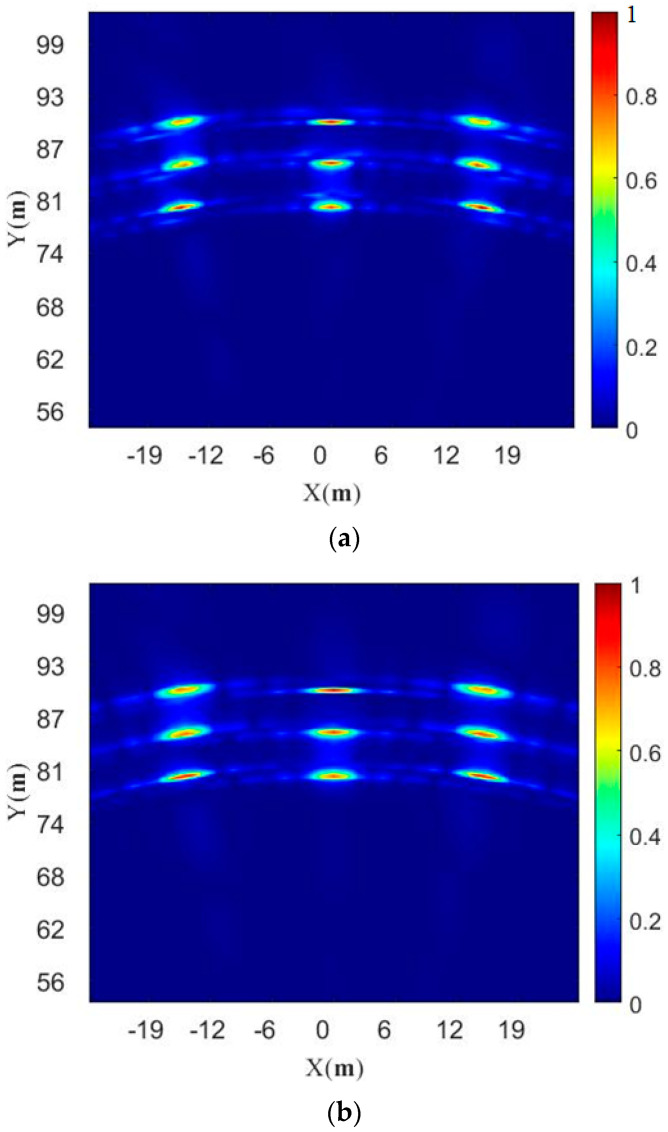
Comparison of imaging results of the 6T8R MIMO radar in different modes; (**a**) Image of targets in random TDM NULA mode; (**b**) Image of targets in TDM ULA mode.

**Figure 8 sensors-23-04756-f008:**
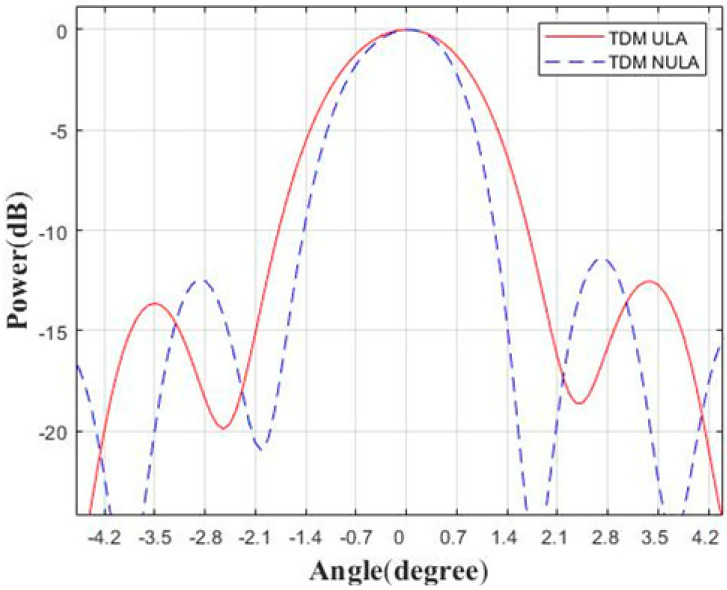
Comparison of point target response in angular direction.

**Figure 9 sensors-23-04756-f009:**
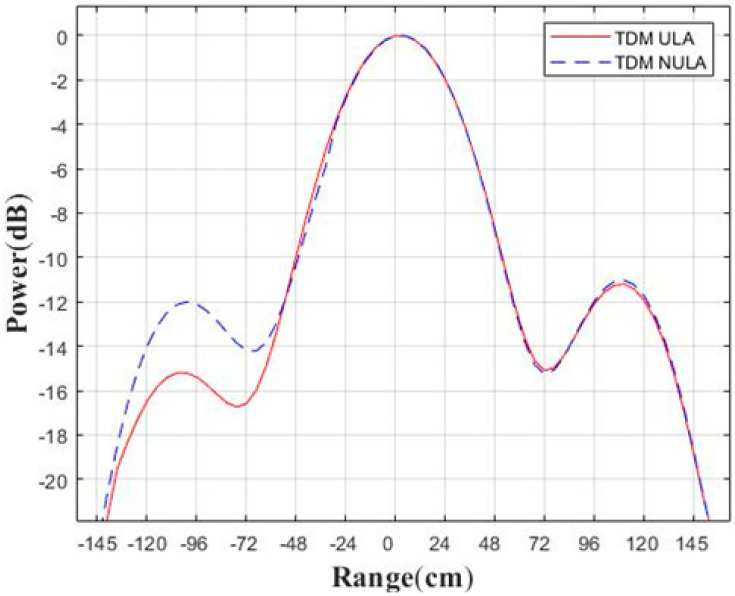
Comparison of point target response in range direction.

**Figure 10 sensors-23-04756-f010:**
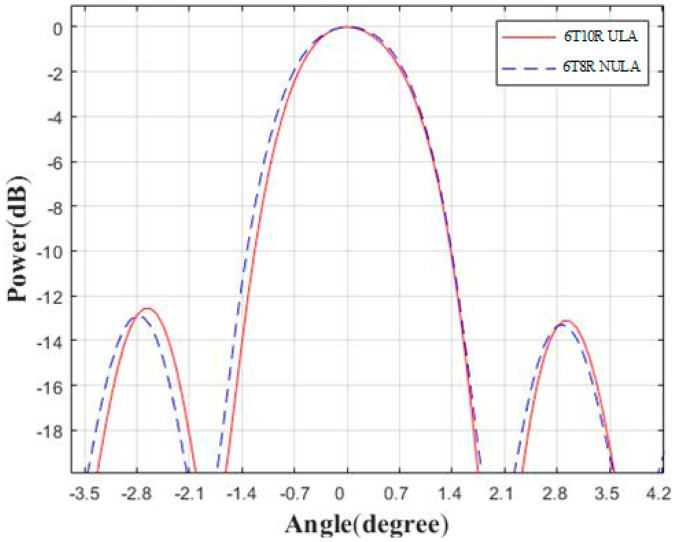
Comparison of angular direction between the 6T10R ULA and 6T8R NULA.

**Figure 11 sensors-23-04756-f011:**
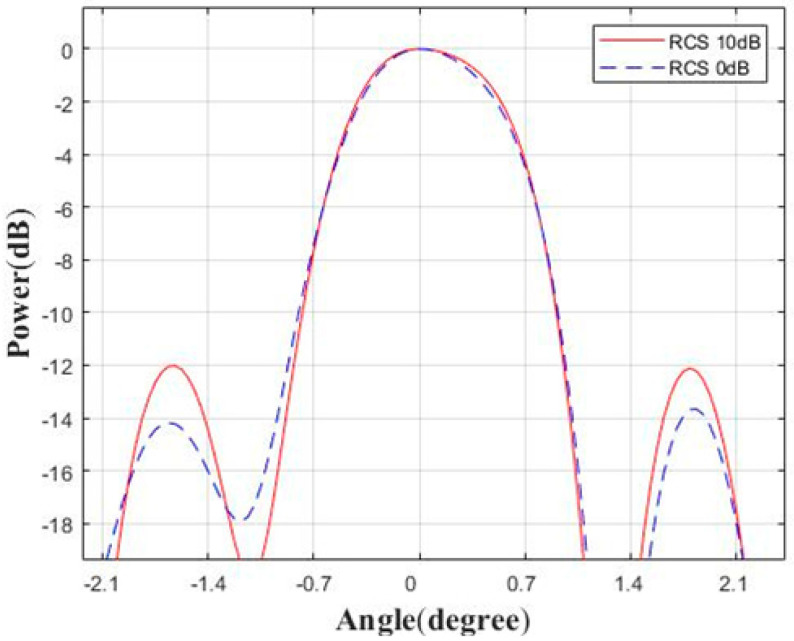
Angular resolution comparison of different RCS targets.

**Figure 12 sensors-23-04756-f012:**
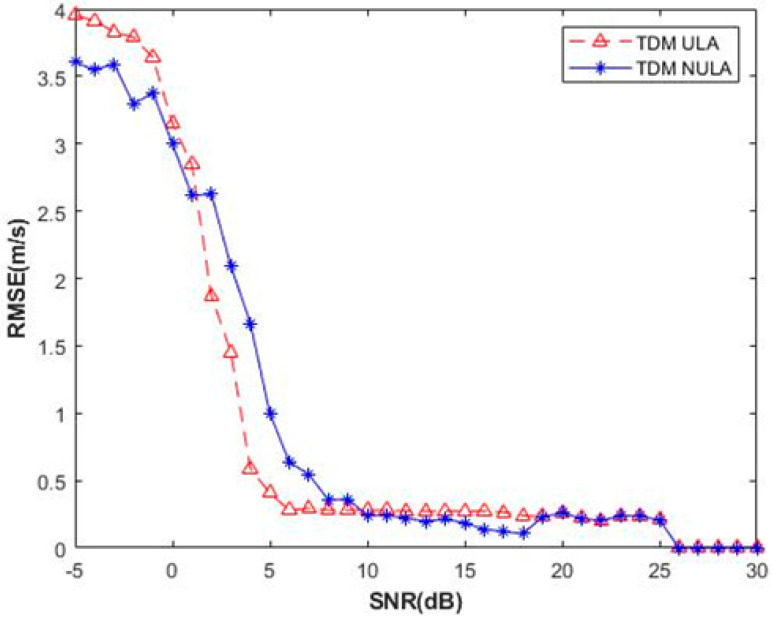
Comparison of the velocity measurement RMSE at different SNR levels.

**Figure 13 sensors-23-04756-f013:**
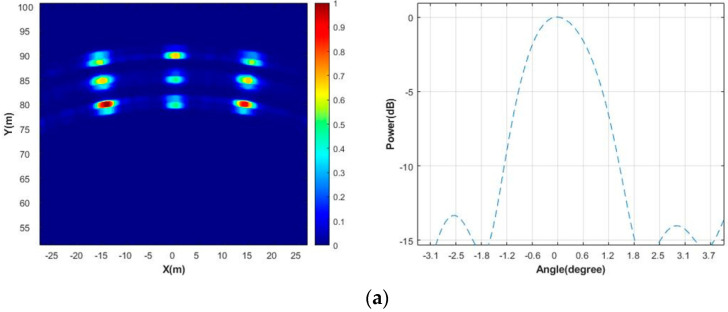
The imaging results of the 6T8R with different array lengths processed by this method; Angular resolution of (**a**) 1.8°; (**b**) 1.6°; (**c**) 1.5°; (**d**) 1.4°; (**e**) 1.3°; (**f**) 1.2°.

**Figure 14 sensors-23-04756-f014:**
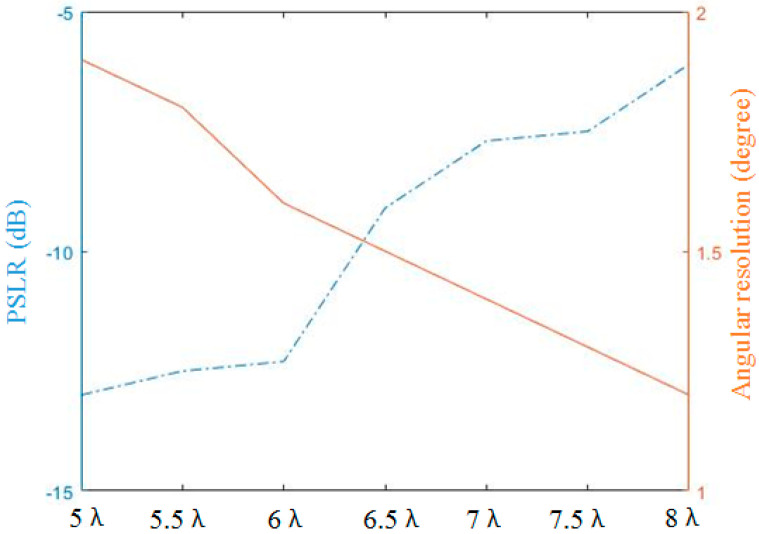
Target response quality at different transmitting antenna intervals.

**Table 1 sensors-23-04756-t001:** Simulation parameters.

Carrier frequency	f0 = 77 GHz
Pulse width	Tp = 50 us
Transmitted bandwidth	300 MHz
Sampling frequency	10 MHz
Number of transmitting antennas	6
Number of receiving antennas	8

**Table 2 sensors-23-04756-t002:** Target response quality statistics.

	Angular Resolution		PSLR	
Transmitting Antenna Interval	TDM–NULA	TDM–ULA	TDM–NULA	TDM–ULA
5λ	1.9°	1.9°	−13.0 dB	−13.0 dB
5.5λ	1.8°	1.7°	−12.5 dB	−14.0 dB
6λ	1.6°	1.6°	−12.3 dB	−14.0 dB
6.5λ	1.5°	1.4°	−9.1 dB	−14.0 dB
7λ	1.4°	1.3°	−7.7 dB	−14.0 dB
7.5λ	1.3°	1.2°	−7.5 dB	−14.0 dB
8λ	1.2°	1.1°	−6.1 dB	−14.0 dB

## Data Availability

Data is unavailable due to privacy policy.

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
