# Peer review of "Random Time Division Multiplexing Based MIMO Radar Processing with Tensor Completion Approach"

_sensors, 2023, doi:10.3390/s23104756_

Round 1

Reviewer 1 Report

In this paper, a novel MIMO radar configuration is proposed, incorporating a non-uniform linear antenna array layout and performing random time division multiplexing operations. This approach yields a third-order sparse receiving tensor signal in the range-Doppler-angle domain. Utilizing tensor completion techniques, the complete signal within the three-order tensor is successfully recovered. Consequently, this enables precise measurements of range, velocity, and angle parameters. However, the paper has the following problems.

1. Please explain the main reasons for the difference between TDM ULA and TDM NULA curves in Figure 8.

2. Why are the main lobes of the two curves in Figure 9 consistent and the side lobes inconsistent?

3. There is a problem with the formula format in the front and rear paragraphs of Section3 formula (21).

4. What is the main reason for random emission combined with NULA?

Reviewer 2 Report

The manuscript describes a random TDM MIMO radar utilizing NULAs. In principal the paper is well written. However, there are no experiments supporting simulations and the concept itself.

Some specific comments which should be addressed in a revised version:

- The tensor completion is optimized by the alternate direction method implying a complex algorithm. This method should be described in much more detail. Especially, boundary conditions, drawbacks, ambiguities of this approach should be pointed out.

- In Fig. 8 ULA and NULA are compared with respect to the resolution in angular direction. The point response of the NULA is narrower, however, sidelobes become somewhat higher. The authors should discuss in how far this is beneficial for automotive applications under realistic conditions.

- In Table 2 the PSLR for different distances of the transmit antennae are simulated and presented. The resolution improves with increasing distance whereas the PSLR degrades. These are simulated values. On the road angular resolution is required between objects with different RCS (van and motorcycles). The authors should discuss in how far their simulations are valid for real scenarios.

- There are no experimental measurements available in this paper which could validate this concept. The lack of realistic experiments certainly reduces the impact of this per se interesting paper.

Reviewer 3 Report

Review of the paper "Random Time Division Multiplexing Based on MIMO Radar Processing with Tensor Completion Approach" by Yuan Zhang et. al.

The paper describes a novel approach to improve the detection capabilities of automotive radar systems. The so-called non-uniform linear array (NULA) exploits non-uniform distances in the receive direction to increase the angular resolution of the system. After a theoretical derivation of the problem, a simulation is conducted in order to validate the improvements.

In general, the paper is well and understandably written. However, several main points need to be addressed before the paper is acceptable for publication: 

-  The extension of the rx antennas (lambda/2 * 8 antennas + lambda/2 = 1.8 cm) is much smaller than for the tx antenna (maximum: 5 * lambda * 6 antennas = 19 cm). If you increase the distance in rx, also the angular resolution should improve. Can you comment on this? Why are the extents limit to rx = 1.8 cm and tx = 19 cm?

- Figure 6: Why is the distance of 1 * lambda between rx5 and rx6? 

- Why is only one scenario with a 1* lambda in the almost center considered. Please extend the paper by an optimization, how an ideal distribution could look like. In addition, please state the boundary conditions of the approach.

- Chapter 4.B: Why is the "new" method not compared to the ULA results?

- Figure 7: It is difficult to see the differences between ULA and NULA. Please generate a difference plot in order to show the improvement. Besides, the font size is very tiny.

Major comments:

- In my understanding, an "experiment" is something to technically prove an approach in the real world. In the present paper, only rather simple simulations are performed. So please remove the word "experiment" throughout the paper, it is miss-leading. 

- Figure 3: The right Figure ("X") is not very intuitive. Where are these "recovered signals" in the grid? Please improve the quality of this plot.

- Page 7, line 218: what is "current industry"? 

- Page 8, line 220: Please replace "Experiment" by "Simulation"

- Page 8, line 236: minimum spacing of 4*lambda. The minimum spacing below is always 5 * lambda. What is the reason for that?

Editorial comments:

- Page 2, Line 66: ..condition of  a  uniform linear... 

- Page 3 line 103: "preliminsries" => What does this word mean?

- Page 3 line 106/107: in this paper ... in this paper... => Please remove the repetition.

Round 2

Reviewer 2 Report

The authors responded to my review in an adequate and scientific way. I am looking forward to see a paper with experimental verification. However, I suggest to publish the revised manuscript.

Author Response

There are no obvious questions about reviewer 2 here, so we did not reply to Reviewer 2. As for the actual experimental verification, we will try our best to implement it in future projects.

Reviewer 3 Report

Second review of the paper "Random Time Division Multiplexing Based on MIMO Radar Processing with Tensor Completion Approach" by Yuan Zhang et. al.

Thank you for the submission of a revised version of the paper. 

Most of my comments were addressed properly. However, I am still not satisfied about the quantitative comparison of the proposed NULA algorithm with the ULA state-of-the-art algorithm.

On page 2, Line 88, the authors write "simulations show that the proposed method can greatly improve the angular resolution". However, when looking in Section 4.A it seems, that the given comparison is not fair: For the ULA approach, the distance in transmit is lambda * 4, while for the NULA approach, the distance is lambda * 5. The comparisons need to be performed with the same initial parameters. Please repeat the simulations / comparison (esp. Figure 8), with lambda * 5 for both ULA and NULA.

In addition, Figure 11 shows only the RMSE for the NULA case. In order to quantify the improvements of the proposed method, also the results for ULA need to be shown in Figure 11.

And please also add the values for ULA with 5*lambda in Table 2.

Maybe I am not that deep into this topic. But generally spoken I think that the improvements in angular resolution seem to come from the increased distance between the antenna elements. This is very obvious and nothing new. The real "new contribution" promised in the abstract or conclusion, the non-uniformity of the linear antenna array is not justified or analyzed properly. Please comment on this.
